# Live Plague Vaccine Development: Past, Present, and Future

**DOI:** 10.3390/vaccines13010066

**Published:** 2025-01-13

**Authors:** Andrey P. Anisimov, Anastasia S. Vagaiskaya, Alexandra S. Trunyakova, Svetlana V. Dentovskaya

**Affiliations:** Laboratory for Plague Microbiology, Especially Dangerous Infections Department, State Research Center for Applied Microbiology and Biotechnology, 142279 Obolensk, Russia; vagaiskaya.anastasiya@gmail.com (A.S.V.); sasha_trunyakova@mail.ru (A.S.T.); dentovskaya@obolensk.org (S.V.D.)

**Keywords:** *Yersinia pestis*, plague vaccine, live vaccine

## Abstract

During the last 100 years, vaccine development has evolved from an empirical approach to one of the more rational vaccine designs where the careful selection of antigens and adjuvants is key to the desired efficacy for challenging pathogens and/or challenging populations. To improve immunogenicity while maintaining a favorable reactogenicity and safety profile, modern vaccine design must consider factors beyond the choice of target antigen alone. With new vaccine technologies currently emerging, it will be possible to custom-design vaccines for optimal efficacy in groups of people with different responses to vaccination. It should be noted that after a fairly long period of overwhelming dominance of papers devoted to subunit plague vaccines, materials devoted to the development of live plague vaccines have increasingly been published. In this review, we present our opinion on reasonable tactics for the development and application of live, safe, and protective human plague vaccines causing an enhanced duration of protection and breadth of action against various virulent strains in vaccination studies representing different ages, genders, and nucleotide polymorphisms of the genes responsible for immune response.

## 1. Introduction

Plague is a poly-host zoonotic infection with multiple modes of transmission caused by the Gram-negative bacterium *Yersinia pestis*. Before the widespread use of vaccines and the era of antibiotic therapy, plague was one of the deadliest diseases in human history, which claimed the lives of over 200 million individuals. The effective use of the first generation of plague vaccines [1,2,3,4,5] together with the successful introduction of antibiotics in plague therapy created a false impression that the devastating plague epidemics remained only in the history of mankind [6]. However, isolation from the natural plague foci of *Y. pestis* strains resistant to all the antibiotics recommended by WHO experts for the treatment of plague [7] and the non-compliance of previously developed and used plague vaccines with recent WHO requirements [8] indicate the need for the development of modern plague vaccines.

At the turn of the 19th and 20th centuries, it was shown that the inoculation of pathogenic microbes, either attenuated or killed, caused protection against the corresponding infection. A single administration of live vaccines, as a rule, ensured the formation of more intense and long-lasting immunity than a similar inactivated vaccine administration. However, live vaccines based on strains with reduced virulence, unlike the inactivated ones, do not exclude the risk of causing an infectious process in immunocompromised individuals. In addition, the reversal of virulence is possible as a result of compensatory mutations in the vaccinal strain. Inactivated vaccines also have their own drawbacks. Therefore, research is ongoing to improve both vaccine forms in order to develop the ideal vaccine combining safety with high protective potency and prolonged immunity [9].

This article briefly reviews the earliest research as well as the current and future approaches to the development of live plague vaccines. Readers are encouraged to consult recent reviews for information on trends in the development of modern plague vaccines and their administration regimens [10,11,12,13,14,15,16]. Here, we provide an overview of how to improve live attenuated plague vaccines based on rational considerations [17].

## 2. Classical Attenuation of *Yersinia pestis*

Attenuation is a decrease in the virulence of strains of pathogenic microbes for their native hosts. It may be a result of serial passages through naive animals of a host species insensitive to infection, immune native hosts, or tissue culture/artificial nutrient media; exposure to bacteriophages, radiation, or chemicals; or lastly due to the site-directed mutagenesis [9,18,19]. Louis Pasteur was the first to generate attenuated strains of pathogens, etiological agents of chicken cholera, anthrax, and rabies, and then used them as the basis for vaccine preparations [20,21]. Currently, live vaccines based on attenuated strains of viruses and bacteria continue to be used and developed [22,23,24,25,26,27]. Both types of whole-cell vaccines have their own advantages and drawbacks. Thus, after a single inoculation, live vaccines provided immunity that was similar to a post-infection one, but their use could cause an infectious process in persons with a reduced immune status. The dead microbes were unable to cause an infectious process; however, numerous doses of inactivated vaccines are generally needed to provide sufficient durable immunity [9].

The empirical and time-consuming classical attenuation of pathogenic bacteria is based on the selection of mutants with reduced virulence formed under the influence of adverse conditions such as long-term passages in animals resistant to the pathogen, in tissue culture, or in nutrient media at altered temperatures and other circumstances inhibiting vital bacterial activity. It is possible to generate attenuated strains much faster by treating microorganisms with mutagens, followed by the selection of clones with reduced virulence [28]. The passaged population is usually genetically heterogeneous and contains both attenuated and virulent microbes. An important step in the generation of attenuated strains is the isolation of clones with the most reduced virulence from the resulting population of microorganisms, followed by the preparation of a genetically homogeneous bacterial population based on such clones. This methodology was used to obtain attenuated vaccine strains of rabies virus (serial passages from rabbit to rabbit [29]), bacillus Calmette and Guérin (BCG) (passaging on a glycerol-bile-potato mixture for 13 years [30]), and yellow fever virus (virus passages in chicken embryos and tissue culture [31]).

The first attempts to attenuate the plague pathogen were made by its discoverer Alexandre Yersin. To do this, he used culturing at elevated temperatures and/or the addition of ethanol at 0.5–5.0% to the nutrient medium. The combination of these approaches made it possible to obtain almost avirulent cultures in two months of passages [4]. In addition, he demonstrated that live attenuated *Y. pestis* protected small laboratory rodents significantly better than killed ones. The protective superiority of live plague vaccines was also shown by other researchers who generated a number of attenuated strains of the plague pathogen [5,32], some of which are presented in Table 1. W. Kolle and R. Otto [33] also showed that infection of mice, rats, guinea pigs or monkeys with slightly attenuated *Y. pestis* mutants can cause the death of some of the animals. However, the surviving animals were protected from death after subsequent infection with a highly virulent strain to a greater extent and for a longer time than those repeatedly vaccinated with killed bacteria. It was also noted that a greater degree of attenuation was usually accompanied by lower immunogenicity [34]. Attenuated strains are also found among closely related microorganisms circulating in nature. In our case, these are representatives of the *microti* subspecies of *Y*. *pestis*, which are, as a rule, avirulent for humans and guinea pigs but highly pathogenic for laboratory mice and their main host, vole [35,36].

At the beginning of the 20th century, it was persuasively shown that vaccine preparations containing exclusively inactivated *Y. pestis* and its culture supernatant could not provide a high level of protection to vaccinated people as well as protect inoculated guinea pigs. There was only a slight increase in resistance to the experimental infection of the latter, while the live vaccine, in very small doses, completely protected cavies from *Y. pestis* infection. Mass vaccination campaigns against plague carried out in Java, Madagascar, and South Africa have shown the high efficacy of live plague vaccines without any accidents among several million human vaccinations [18,41]. It should be noted that the vaccine strains used were derived from different parental strains and attenuated using different procedures based on different theoretical backgrounds (Table 1).

Five conclusions can be drawn summarizing the results of this stage of research on the development of a plague vaccine:Live plague vaccines are superior to the killed ones in terms of intensity and duration of induced immunity.The simultaneous use of two independently attenuated *Y. pestis* strains potentiates the protective potency of each of them.Immunizing properties are dependent on more than one antigen. Different antigens fluctuate in their ability to induce immunity in diverse mammalian species. The immunogenicity of plague vaccines should be evaluated in mice, guinea pigs, and monkeys.Vaccines based on live attenuated *Y. pestis* strains may be fatal to some immunized mammal individuals with weakened immunity or metabolic disorders.A mixed vaccine including the strains with special efficacy in rats and in guinea pigs protects both animal species better than the monocomponent ones.

## 3. Discovery of *Y. pestis* Virulence Determinants and Subcellular Fractions Responsible for Pathogenicity and/or Immunogenicity

The disadvantages of killed and live whole-cell plague vaccines have initiated the search for bacterial subcellular components that induce the development of a protective immune response. H. Schütze [42,43,44] described two antigens that are components of *Y. pestis*. The antiphagocytic envelope antigen (envelope substance, capsular antigen, fraction I, F1, antigen A, antigen 3 [38], and Caf1 [45]) was contained in the gelatinous capsule formed at 37 °C, while the water-insoluble somatic antigen (basic somatic antigen (BaSoAn), “residue antigen” [32,46]) was a part of the somatic portion of the bacterium. The envelope antigen possessed significant prophylactic power for rats, mice, and monkeys [32], while the water-insoluble “residue” antigen remaining after the saline extraction of acetone-treated bacteria [38,47,48] and consisting of 70–75% of the whole bacterium [49] was protective for guinea pigs and monkeys [32,50].

In 1956, T.W. Burrows described the virulence antigen Vi [51], and later renamed it as V [38]. Like F1, it was also antiphagocytic, but when the culture temperature changed from 28 °C to 37 °C, its activity emerged several hours earlier than the comparable activity of F1 [38]. The immunization of rabbits induced higher titers of antibodies to the V antigen compared with similar indicators in guinea pigs and mice. Furthermore, immune rabbit serum provided passive protection of mice from infection with the wild-type strain of *Y. pestis*. Guinea pig immune serum was not protective.

The summarized results of 60 years of research from several laboratories on *Y. pestis* strains differing in virulence allowed T.W. Burrows to propose the concept of virulence determinants. The absence of one or more of these biomolecules or properties in a pathogen is accompanied by its attenuation [38,48,52]. T.W. Burrows perfectly understood the limitations of the concept he put forward based on the study of non-isogenic strains and experimentally unsupported assumptions. Direct proof of the importance of any determinant in the pathogenesis of plague would require demonstrating that a “wild-type” strain that has mutated in the gene encoding that virulence determinant would decrease its virulence, and the complementation of the non-functioning gene would restore virulence to its original high level (Table 2). In an attempt to prove this direct evidence, he observed several challenges including the following:The possibility of losing not only the determinant, but also some unidentified genes;The absence at that time of tools for genetic exchange using models that would allow complementation. As for the evidence that a particular factor plays a minor role in virulence, it suffices to show that its loss does not reduce the virulence of the pathogen [48]. Another problem he mentioned is that the overwhelming majority of researchers did not use standard operating procedures when assessing immunity to plague, which prevents a reliable comparative assessment of the results obtained in different laboratories using different vaccine strains and different test-infecting strains administered at different doses. Unfortunately, the latter situation has so far remained the same [53].

**Table 2 vaccines-13-00066-t002:** Live attenuated *Y. pestis* strains used as vaccines against plague and generated using molecular bacteriology methods [54,55,56,57,58,59,60,61,62].

*Y. pestis* Strain	Virulence for (Approximate LD_50_ Values in cfu)		Protective Potency for	
	Mice	Guinea Pigs	Mice	Guinea Pigs
Subcutaneous Challenge				
Biovar Microtus strain 201	Avirulent to humans or primates		+	ND
Wild-type subspecies *pestis*	<10	<10	+	+
Δ*pgm*	>10^8^	>1.5 × 10^10^	+	+
pPCP^−^	<10 to >10^8^	<10 to >10^8^	ND	ND
pYV^−^	>1.0 × 10^8^	>1.5 × 10^10^	−	ND
pMT1^−^	<10	<10	+ **	
Δ*pgm* pPCP^−^	>1.0 × 10^8^	ND	+	ND
Δ*pla*	<10 to >10^8^	<10	+	ND
Δ*nlpD*	>10^7^	>1.5 × 10^10^	+	−
Δ*yopH*	>10^7^	ND	±	ND
Δ*dam*	2.3 × 10^3^	ND	+	ND
Δ*relA* Δ*spoT*	5.8 × 10^5^	ND	+	ND
Δ*crp*	>3 × 10^7^	ND	+	ND
Δ*yscB*	>10^6^	ND	+	ND
Δ*glnALG*	>10^5^	>10^7^	+	+
Δ*metQ*	>10^5^	>10^8^	−	−
Δ*ailC*				ND
Δ*lpp* Δ*msbB* Δ*ail*	>2 × 10^6^	ND	+	ND
Δ*lpp* Δ*msbB*::*ailL2*	>2 × 10^6^	ND	+	ND
Δ*ypo2720*-*2733*Δ*hcp3*	60% *	ND	+	ND
Δ*vasK*Δ*hcp6*	60% *	ND	±	ND
∆*yscN*	>3.2 × 10^7^	ND	+	ND
∆*surA*	>2.1 × 10^5^	ND	+	ND
**intranasal challenge**				
Δ*lpp* Δ*msbB* Δ*pla*	>2 × 10^6^	ND	+	ND
Δ*smpB-ssrA*	>10^6^	ND	+	ND

*, % animal survival in pneumonic plague challenge; ND, no data. **, +—protective from F1-negative strains.

Further studies of plague pathogenesis and immunogenesis in molecular biology, genomic, and post-genomic eras [60,61,62,63,64,65,66], based on Koch’s postulates modernized by S. Falkow [67], allow scientists to identify a significant number of additional pathogenicity factors necessary for the full virulence (Pla, PsaA, Ymt, Ail, LPS, etc.). Among them, there were no antigens comparable in protectiveness to the F1 and V antigens [68]. This was the reason for a majority of researchers to switch their focus to developing various versions of low-component subunit vaccines based on F1 and V antigens. In principle, vaccines based on nucleic acids [69,70], viral or bacterial vectors [71,72], plant-derived plague vaccine [73], nanoparticles [74], and microencapsulated preparations [75,76] can also be included in the group of subunit vaccines, since they all induce an immune response to only one to three target immunodominant antigens. However, it would be useful to acknowledge that (i) subunit vaccines have a place in a non-endemic setting, (ii) they may be especially useful for annual revaccinations after initial prime immunization, and (iii) their development has helped to define factors essential for protection.

The main advantage of subunit vaccines is their non-infectious nature. Other benefits include low reactogenicity and relative ease of authentication control. However, according to a number of researchers, they have no or limited innate defense triggers and do not guarantee the formation of a good immune memory. The incorporation of Toll-like receptor agonists in subunit vaccines addresses this problem. The immunogenicity of subunit vaccines can be increased through the use of adjuvants (for example, adsorption on aluminum hydroxide) [77].

Different mammalian species, subspecies, and even populations, including diverse racial/ethnic groups of humans, vary in their susceptibility to plague [5,46,55,78,79,80,81,82,83,84] and other severe bacterial infections [85]. They also differ in their ability to respond to different antigens of bacterial pathogens, causing the humoral and/or cellular species-specific immune response [84,86]. The reason for this lies in the differences between mouse and human immunology [87,88,89].

In addition, age- and sex-related differences in the immunopathogenesis of infectious diseases have also been recorded. Susceptibility to and mortality from many infections is higher among men, and the immune response to various types of vaccinations is usually stronger in women, which is likely due to stronger humoral responses. The major sex steroid hormones have opposing effects on the cells of both the adaptive and innate immune systems: estradiol generally enhances, while testosterone suppresses the response [90]. Recently, sex-related differences in plague vaccine efficacy and the immunological factors that differ between male and female BALB/c mice were shown. The protectivity of various experimental plague vaccines was higher in the female mice. Males had a higher bacterial load and differed in cytokine expression patterns and serum antibody titers. The synergy between vaccination and antibiotic treatment repeatedly observed in the female mice was absent in the male mice [91]. Taking into account the aforementioned and, above all, the fact that the vast majority of experiments assessing the protectiveness of candidate plague vaccines have been carried out using a mouse model, their results must be extrapolated to humans with great caution.

In addition to differences in the interspecies-specific and intraspecies-specific host response to *Y. pestis* antigens, it is also necessary to take into account *Y. pestis* antigenic variability. To evade host immune defenses, pathogens evolve by producing isoforms of immunodominant antigens, the bacterial factors absolutely necessary for the virulence of the pathogen [92,93], or reversibly [94,95] or even completely [96] losing the ability to produce these antigens if their loss does not reduce virulence.

In closing this section, the following should be emphasized:Two-component subunit plague vaccines based on the F1 and V antigens provide good protection in mice [97,98], but their efficacy varies significantly among non-human primate species, and total antibody titers to plague antigens do not always correlate with protection [10,14,99].These antigens induce primarily antibody-mediated immune responses in humans [100,101,102] with widely varying antibody titers and induce weak cellular responses [103], which play a decisive role in human immune protection against plague.Such low-component vaccines will be of little effectiveness against infection by strains producing serologically distinct isoforms of V [92] and F1 [93] antigens or by virulent mutants that fail to form an envelope from the F1 capsular antigen [104].

Taking into account the above considerations, it is hoped that a whole-cell vaccine, containing a complete or almost complete set of pathogen antigens, can provide protection against both “classical” and antigen-altered variants.

## 4. Recent Progress in the Development and Use of Live Plague Vaccines

### 4.1. Current Plague Vaccination

Recent progress in high-throughput “omics” methodologies and the public accessibility of the complete genome sequences of hundreds of strains of many species of pathogenic bacteria have fundamentally changed the scope for picking out new vaccine candidates. In the course of the development of plague vaccines, a number of promising vaccine strains were generated, some of which were successfully used as a live vaccine for large-scale immunization involving several million people (more detailed information about these studies can be found in reviews [10,11,12,13,14,15,68,105,106,107,108,109]), but only one of them, based on the NIIEG line of the EV76 strain, is currently approved for the vaccination of people in a limited number of countries.

In the Soviet Union, more than 10,000,000 people were vaccinated with the live vaccine based on the NIIEG line of the EV strain [110]. An analysis of the results of vaccination with the live vaccine based on the EV line Saigon of the population of six provinces of South Vietnam involving 2,089,388 people indicates that vaccination did not fundamentally affect the reduction in morbidity among those vaccinated, but the course of the disease was milder and complications in the form of secondary pneumonia were less frequent [111,112]. According to N.I. Nikolaev, in 1945, during the period of the exacerbation of the epidemiological situation in Inner Mongolia, the use of the live plague vaccine made it possible to reduce the incidence in the vaccinated group (0.25 per 1000 people) compared to the unvaccinated (28.8 per 1000 people) by 100 times [110]. A high efficiency has also been demonstrated with the use of the *Y. pestis* EV vaccine strain in the Belgian Congo and South Africa [113,114].

Currently, a live plague vaccine based on the NIIEG line of the EV76 strain is used annually in quantities of several tens of thousands of doses in Russia and Kazakhstan to immunize the staff of anti-plague laboratories and the population living in enzootic areas. Over the last five years, 90,822 people have been vaccinated against plague in the Altai Republic [110,115].

It is obvious that the live vaccine based on the NIIEG line of the EV strain will continue to be used for specific plague prevention until a new drug is developed that is superior to the existing one [116] in at least some of the parameters noted in the WHO target product profile for plague vaccines [8].

### 4.2. Criteria for Selecting Candidate Vaccine Strains

The development of any vaccine candidate follows general rules, but each development has its own specifics depending on the type of vaccine (live/killed/subunit/DNA/peptide), pathogen–host interaction at the population level, target product profile, target population, and availability of an existing vaccine. To include a vaccine candidate in the drug development cycle, it must meet a number of criteria. However, international standard operating procedures for assessing plague immunity are still lacking, which does not allow for a reliable comparative assessment of results obtained in different laboratories using different candidate vaccine strains and different test-infecting strains administered at different doses [19,117,118].

In Russia, all trials of attenuated plague strains that are promising as vaccine strains are conducted in comparison with the reference *Y. pestis* vaccine strain EV line NIIEG [119]. In abbreviated form and in English, these requirements are discussed in the article [98]. According to this publication, candidate vaccine strains of *Y: pestis* must match or exceed the reference vaccine strain EV in terms of immunogenicity. In addition, it should match the control strain in terms of harmlessness and reactogenicity or be safer, and may differ from the EV strain in some insignificant characteristics but still maintain it as a representative of the species of *Y. pestis*.

The candidate vaccine strain being studied must meet the following requirements:It must be lysed by the plague diagnostic bacteriophage L-413C [120];It must be typical in its cultural and morphological properties [121];The F1 titer of the studied strain must not be less than the similar indicator obtained with the culture of the control strain *Y. pestis* EV grown under similar conditions;The proportion of calcium-independent mutants in the population of *Y. pestis* cultures [122] passaged through the body of laboratory animals and not subjected to long-term storage or any physical impacts must not exceed 0.3%;It must not be inferior to the control strain in fibrinolysin-coagulase activity [123];The studied and control strains must not have the ability to pigment sorption (pigmentation) [124];The studied vaccine strains, similar to the reference strain EV, should have three bands of plasmid DNA on the electropherogram, corresponding to pMT1 (60 MDa), pYV (47 MDa), and pPCP (6 MDa) [121].

When reading these requirements, it seems that they were developed to obtain strains that are closest in properties to the EV vaccine strain. In fact, attenuation can be carried out not only by eliminating the *pgm* locus from the genome [124], but also by knocking out other genes encoding pathogenicity factors [68]. A change in the structure of lipopolysaccharide during attenuation due to the shortening of the core oligosaccharide [125] should prevent the interaction of the diagnostic phage with the surface of the bacterial cell [126]. Cultural and morphological properties can most likely change when editing the genes encoding the surface structures of the bacterium [127]. The production of plasminogen activator does not contribute to the development of intense immunity, and it is advisable to remove the gene encoding it from the genome [55,128].

There is no doubt that despite the variety of candidate vaccine strains, all of them must meet at least the following criteria:Absolute safety;High vaccine protective efficacy.

The methods used to test these criteria should be standardized and their number reasonably minimized.

The most accessible way to assess specific immunity in vaccinated individuals is to determine the level of specific antibodies, but the presence of specific antibodies even to immunodominant antigens F1 and/or V does not completely and not in all mammals correlate with the host’s protection from infection [129], since the leading role in the formation of anti-plague immunity belongs to cellular factors of the immune system [130].

### 4.3. Yersinia pestis Natural Strains Selectively Virulent or Non-Pathogenic (Conditionally Pathogenic) for Humans

*Y. pestis* species includes a group of strains (pestoides group, vole’s strains, bv. Microtus, subsp. *microti*) circulating in populations of various species of voles or Mongolian pikas. These strains are highly virulent in their natural hosts and laboratory mice, but they are generally avirulent in larger mammals, including humans [35,46]. In fact, these strains are already candidates for vaccine strains.

The evaluation of the protective efficacy of the subsp. *microti* strain 201 showed that this strain corresponded to the vaccine strain EV in terms of the degree of attenuation upon subcutaneous administration, the ability to induce the synthesis of antibodies to F1, and the protection of rhesus macaques from infection with *Y. pestis* [131]. It would seem that the problem of a live plague vaccine in terms of obtaining candidate vaccine strains has been solved since natural plague foci (including those in which vole strains, conditionally pathogenic for humans, circulate) are a virtually inexhaustible source of *Y. pestis* isolates. However, there are several “buts”:Vole strains are characterized by the polymorphism of amino acid sequences of a number of proteins (including F1 [93] and V antigens [92]); several laboratory-confirmed cases of the isolation of *microti* strains from humans have been described [46];The possibility of increasing subcutaneous virulence for guinea pigs to levels similar to that of the strains of the main subspecies was shown through testicular passages [132].

So, in order to be confident in the safety of vole strains in the event of their use as vaccines, it is necessary to introduce into their genome one or two additional alternative attenuating mutations.

### 4.4. Selective Protective Potency of Yersinia pestis

The majority of plague experts believe that differences in the clinical form and severity of plague infection are not determined by age, gender, or ethnicity, but are due only to differences in exposure to infection [133,134]. Nevertheless, a number of data indicate that the different severity of the infection and the intensity of immunity is determined by the genotype (ethnicity) of the host [135,136,137] in interaction with the specific phylogenetic group (subspecies) of the pathogen [35].

The effectiveness of the vaccine-induced immune response is determined by the interactions of immunocompetent cells as well as by the duration of antigen circulation in the vaccinated person. Polymorphism of the genes responsible for the formation of an immune response may be the reason for an insufficient intensity of the immune response. Thus, the immunization of volunteers with a subunit plague vaccine induced the production of antibodies to F1 and/or V antigen in 67% of them in titers not less than the threshold level. However, 33% of the vaccinated people remained seronegative. A study of 20 single nucleotide polymorphisms in 14 immune response genes revealed links between homozygosity or heterozygosity for the allele variants of these genes and susceptibility or insensitivity to immunization with different antigens [137].

## 5. Strategies Aimed at Increasing Genetic Stability

Currently, in countries where there are no licensed plague vaccines, the main efforts of researchers are aimed at developing new remedies that meet the WHO requirements [8]. In Russia and Kazakhstan, countries of the former Soviet Union, where the live plague vaccine based on the EV vaccine strain continues to be constantly used to immunize tens of thousands of people annually, most research has been aimed at preserving the properties of the initial vaccine strain in terms of safety and protective activity during the prolonged storage of a stock culture. To a large extent, this problem was solved by using freeze-drying and storing dry vaccines in low-temperature refrigerators [138].

The accumulation of various mutations arising during the passage and laboratory storage of EV76 strain lines maintained in laboratories in different countries [138] was accompanied by a decrease in the protective capacity of most EV76 strain lineages [139].

This problem can be solved in several ways as specified below:Comparative tests were carried out on all lines of the EV vaccine strain supported in the USSR. The freeze-dried NIIEG lineage retained a maximum protective activity and was grown and packaged as stock cultures for the subsequent production of live plague vaccine [138]. Currently, in countries where there are no licensed plague vaccines, the main efforts of researchers are aimed at developing new remedies that meet the WHO requirements [8].The animalization of the vaccine strain is carried out with the aim of purifying its population from mutants that have a reduced viability.

The “animalization” of the *Y. pestis* EV vaccine strain was carried out by its intravenous injection into a rabbit and the subsequent (after 3–4 h) isolation of the vaccine strain from its internal organs. A comparative assessment of the original and animalized bacterial cultures showed a significantly greater number of viable microbes in the preparations of the latter, combined with a significantly greater degree of protection when infected with a virulent strain [140].

Another option for animalizing the vaccine strain involves three testicular passages in guinea pigs [141].

3.Stabilization of the genome of the vaccine strain by genetic engineering methods is also possible. Recombinase RecA is responsible for most acts of homologous genetic recombination in bacteria [142,143]. To overcome an unwanted homologous recombination that destabilizes the genome of the vaccine strains of various bacterial species, researchers created *recA* deletion mutants, since RecA is mainly involved in recombination in bacteria [144,145].

The deletion of the BCG *recA* gene leads to a complete loss of recombination between homologous sequences located on the chromosomes, as well as between sequences located on the plasmid and on the chromosomes. Mutant BCG Δ*recA* was as effective as the wild-type one in protecting mice from intravenous challenge with virulent *Mycobacterium tuberculosis*, indicating that the loss of the SOS-mediated DNA repair mechanism does not compromise the immunological properties of BCG. The availability of genetically stable, fully immunogenic BCG is important for the future development of BCG as a live vaccine [146].

The construction of a *recA* knockout mutant was successfully used to stabilize the genome of *Francisella tularensis* subsp. *holarctica* vaccine strain 15 lineage NIIEG. Compared to the parent strain, the constructed 15/23-1Δ*recA* mutant multiplied in macrophage-like J774A.1 cells 8-10 times more slowly. BALB/c mice responded to immunization with strain 15/23-1Δ*recA* with less weight loss (~2%) than to strain 15 (~14%). *F. tularensis* strain 15/23-1Δ*recA*, which has reduced reactogenicity, has been proposed as the basis for the creation of a stable and safe live tularemia vaccine [147]. It makes sense to try to stabilize the genome of candidate *Y. pestis* vaccine strains in a similar way.

## 6. Synergy of Action of Multi-Strain Vaccines

As noted above, the synergistic effect of combined immunization with two differently attenuated *Y. pestis* strains resulted in reliable protection against plague [4,148] caused by both wild-type and nonencapsulated *Y. pestis* strains. Such double-strain vaccines include both bacteria covered with an envelope formed from the F1 antigen and bacterial cells whose outer membrane antigens (LcrV, YopD, YpkA, etc.) are open for interaction with the host immune system.

## 7. Future of Plague Live Vaccines

The need to improve an existing commercial plague vaccine and/or develop new effective and safe vaccines can be realized in several directions of research. Regardless of the chosen approach, it is desirable that a vaccine being developed presents to the immune system multiple antigens and induces both humoral and cellular immune responses, since both arms of the immune response appear to be critical in anti-plague vaccine strategies. Unfortunately, vaccines not only protect against infectious diseases but also can cause both general and local post-vaccination adverse reactions [149,150], one of the causes of which may be allergies [151]. These reactions are most often associated with the non-microbial components of the vaccine, but in many cases, the specific ingredient causing the reaction cannot be definitively identified [149]. A high level in the blood of IgE, which is responsible for the formation of allergic reactions, was noted in people who were repeatedly vaccinated against plague [152]. A direct dependence of the frequency of allergic diseases on the frequency of vaccination of the employees of anti-plague institutions has also been revealed [153]. An in silico analysis of 3256 proteins made it possible to identify 170 (5.22%) probable allergenic proteins of the vaccine strain *Y. pestis* EV line NIIEG. A total of 38 allergenic proteins belonging to the extracellular and outer membrane groups have been identified as the most promising targets for the creation of hypoallergenic vaccines by removing the genes encoding them from the genome of the vaccine strain EV.

To further optimize the EV vaccine, one research group has proposed the deletion of the *pla* gene encoding the plasminogen activator [128]. The product of this gene is not significantly protective but is one of the main pathogenicity factors of *Y. pestis* [123]. It is likely that the deletion of other ballast components may become one of the directions for improving both the existing and newly constructed live plague vaccines.

Taking into account the poly-host nature, the several clinical forms of plague, and the multi-modes of its transmission, it can be assumed that in different mammalian species, different antigens of the plague pathogen can be protective. Research into the creation of combined plague vaccines based on the derivatives of one or several *Y. pestis* strains, attenuated in various ways, which have shown a different breadth of protection against various virulent strains in a model of one or several species of laboratory animals, was initiated in the first half of the XX century. The evaluation of the efficacy of a candidate plague vaccine in several animal species (mice, rats, guinea pigs, and monkeys) with various species differences in the immune system responses gives reason to hope that the vaccine will be effective in humans. However, caution should be exercised when proposing multiple animal models as this complicates the comparison of data from different laboratories.

Live vaccines #46-S, M # 74, or MP-40 (F1^−^), when administered together with EV in the form of a two-component mixture, significantly potentiated each other’s protective activity [4]. The bivalent vaccine composed of *Y. pestis* strains 1 and 17 competed in protection with the vaccine based on the EV strain, but turned out to be more reactogenic [11,154]. More recently, it was shown that immunization with a precisely attenuated *Y. pestis* strain accompanied by the administration of a protein subunit vaccine was significantly more protective than the administration of individual preparations [148].

In a recent study, adenoviral vector-based (genes encoding YscF, F1, and V) and live-attenuated Δ*lpp*Δ*msbB*Δ*ail Y. pestis* CO92 derivative vaccines were successfully used in a mixture, with each vaccine inducing a distinct cellular immune response [155].

*Y. pestis* Δ*nlpD* mutant has recently been shown to be able to exceed the defensive potency of the vaccine strain EV [156]. The comparative testing of a set of Δ*nlpD* strains of different origins confirmed that the immunization of mice with Δ*nlpD* mutants induces immunity 10^5^ times more potent than the one induced by the administration of the EV vaccine strain. Simultaneously, NlpD-deficient bacteria failed to protect guinea pigs in the case of a subcutaneous challenge with *Y. pestis*, inducing a 10^6^ times less potent protection compared with that conferred by immunization with the EV vaccine strain [127]. It would be advisable to evaluate the protective activity of a mixture of Δ*nlpD* and Δ*pgm* strains in these two animal species.

Recently, a set of technologies based on regulated-delayed gene expression has been developed. On the one hand, this approach allows pathogens to remain in the body of a vaccinated person in quantities and for a time sufficient to form long-term and intensive immunity. On the other hand, this method guarantees the impossibility of the infection process. A review of those technologies is presented in [157], but we would recommend the readers to follow the original publications [158,159,160,161,162,163,164,165,166] which deserve it.

## 8. Conclusions

Vaccination against plague remains a challenging issue not only due to the lack of globally accepted licensed vaccines but also due to the absence of generally accepted methods for comparing their safety and efficacy. However, the plague is endemic in large populated areas of Eurasia, Africa, and the Americas, which maintains its pandemic potential. It probably makes sense to continue research into the development of alternative strategies for specific plague prevention based on both live and inactivated vaccines and/or their combinations.

In our opinion, in regions endemic for plague, the most promising is prime immunization with a live vaccine based on two or three attenuated strains, at least one of which should be F1-negative. The candidate vaccine strains of the plague pathogen should not be fatal to guinea pigs in doses up to 1.5 × 10^10^ cfu, and to mice in doses up to 10^7^ cfu. The strains should have knockout mutations or, better, a controlled delayed attenuation.

Then, revaccination with subunit vaccines based on F1 and V antigens in combination with bacterial ghosts or outer membrane vesicles should be carried out annually. This approach is also acceptable for the personnel of plague laboratories.

If a candidate vaccine is generated that is superior to the existing nationally licensed ones, it will be necessary to seek permission for its use in individual natural plague foci.

## Figures and Tables

**Table 1 vaccines-13-00066-t001:** The first attenuated *Y. pestis* strains used as vaccines.

*Y. pestis* Strain	Method of Generation	Presence of Major Immunodominant Antigens and *pgm* Locus	Protective Efficacy	Harmless to		Author	Reference
				Animals	Humans		
MaV	ND	ND	ND	ND	Vaccine-related casualties were not described among 1101 vaccinated people.	P. Strong	[37]
AMP	Treatment with a bacteriophage	ND	Inferior to EV strain	Harmless in doses up to 24 × 10^9^ CFU for mice and gophers, but some guinea pigs died.	Harmless when administered subcutaneously or inhaled in doses up to 1.5 × 10^9^ CFU (more than two thousand people were immunized).	M.P. Pokrovskaya	[4]
ZhV	Treatment with a bacteriophage	ND	Eventually lost its immunogenicity and became inferior to EV strain	Doses of 50 × 10^9^ CFU caused the death of individual guinea pigs.	ND	N.N. Zhukov-Verezhnikov	[4]
#46-S *	Treatment with a bacteriophage	ND	Inferior to EV strain	ND	ND	E.I. Korobkova	[4]
M # 74 *	20 years of reseeding on artificial media	ND	Equivalent to EV strain	Avirulent for mice, guinea pigs, and rabbits in doses up to15 × 10^9^ CFU.	Avirulent for human volunteers.	N.N. Zhukov-Verezhnikov, T.D. Fadeeva, A.P. Yashchuk	[4]
Tjiwidej	After rat passage followed by 4-month maintenance on agar-serum medium at 5 °C, the strain was found to be avirulent.	Pgm^+^ V^−^ F1^+^	Protects rats better and guinea pigs worse than the EV strain	Avirulent for guinea pigs and rats; LD_50_ for mice is 1.5 × 10^8^ CFU.	Extensively used as a live vaccine in human plague prophylaxis.	L. Otten	[2,37,38]
MP23	Tjiwidej derivative subjected to X- or ultraviolet radiation. Irradiated samples after storage for 24 h. at 5 °C on tryptic meat agar were incubated for 16 h. at 37 °C, and the resulting organisms were injected intraperitoneally into 20–50 mice (1 × 10^7^ cells per mouse).	V^−^	Highly immunogenic for guinea pigs and macaques	Virulent for mice, but avirulent for guinea pigs and macaques. About 50% of vervets and 100% of langurs succumbed to the vaccination.	ND	T. Burrows, G. Bacon	[37]
MP-40 *	Isolated from ground squirrel infected during hibernation followed by passage through cavy immunized with 300 × 10^6^ CFU of *Y. pestis* vaccine strain EV and up to 20 subsequent passages at 40 °C through broth with 10% ethanol.	F1^−^	ND	ND	ND	Kasuga	[4]
Harbin	ND	Δ*pgm* F1^+^	ND	ND	ND	ND	[34,39,40]
EV	Five years of monthly reseeding (total 76) on solid artificial media at 18–20 °C.	Δ*pgm* F1^+^V^+^	Highly immunogenic for mice, guinea pigs, and monkeys	Avirulent for guinea pigs and rabbits.	Since 1932, more than 10 million people have been safely vaccinated without fatal plague cases due to immunization.	G. Girard, J. Robic	[3]

ND—no data. * vaccines #46-S, M # 74, or MP-40 (F1^−^), when administered together with EV in the form of a two-component mixture, significantly potentiated each other’s protective activity [4].

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
