# Peer review of "Live Plague Vaccine Development: Past, Present, and Future"

_vaccines, 2025, doi:10.3390/vaccines13010066_

Round 1

Reviewer 1 Report

Comments and Suggestions for Authors

This is a well-considered review and definitely a useful addition to the field, from this  respected author. Overall I enjoyed reading it and found it informative. I have some comments to make which are intended  to be helpful.

The review is written by someone who lives in a region endemic for plague and therefore appreciated the need for highly effective vaccine to curtail the real possibility of out breaks. 

However, this is not the universal situation and I would  argue that regulatory bodies in the West have had a more cautious approach to licensing live vaccines for plague, so far at least. So whilst I accept that 'there has been an overwhelming dominance of sub-unit vaccine' reports, (Abstract lines 13-15)these were driven by the requirements from the regulators in the West (predominantly the FDA) to 'better define' vaccines, above all with respect to absolute safety. Indeed a microencapsulated  sub-unit vaccine for plague has been licensed in Russia (referenced) but there is no elaboration of its utility. I suggest it would be useful acknowledge that    a)subunit vaccines have a place in a non-endemic setting and b) that their development has helped to define factors essential for protection. Line 179 would seem an appropriate place to make such amendments.

The authors bemoan the ack of consistency in standard operating procedures to date  in plague vaccine research, with which I agree and which via WHO fora in recent years is being addressed, but why do they persist n calling the pYV/pCD1 plasmid  pCAD; this is a minor point, but it did rather irritate this reviewer.

Lines 219-220: ref 98 is quoted to support the claim that 'antibody [to plague antigens] does not correlate with protection.' This reference does not say that, rather  it acknowledges that total antibody does not correlate, but that functional antibody i.e. in this case, anti-V antibody which is neutralising, does correlate.    Indeed, labs around the world are starting to evaluate the development of neutralising antibody by competitive ELISA, with the hope that this may become a surrogate marker of efficacy. This is not to dismiss the role of Cell-mediated immunity, as I agree that both functional antibody and CMI are important in protection.   

Minor points are; line 91: a word is missing after 'attenuated'. Also please be cautious in suggesting multiple animal models are required to demonstrate vaccine efficacy (mouse, rat, guinea pig, macaque) as this adds to the difficulty of comparing data across labs/ultimately achieving a vaccine for man (with the caveat that early stage research may need several models to assess a candidate.

Finally I would like to have seen a conclusions/recommendation section as the bulk of the review is on EV76 and derivatives, with a late mention of a LAV plus subunit combination and other approaches to LAV's such ad the delayed in vivo regulation. What is the bottom line opinion from this group of well-respected authors?    

Author Response

Dear reviewer, thank you very much for your thorough analysis and high evaluation of our work.

Comment 1: This is a well-considered review and definitely a useful addition to the field, from this respected author. Overall, I enjoyed reading it and found it informative. I have some comments to make which are intended to be helpful.

Response 1: Thank you for your high appreciation of our manuscript.

Comment 2: The review is written by someone who lives in a region endemic for plague and therefore appreciated the need for highly effective vaccine to curtail the real possibility of outbreaks. 

Response 2: Indeed, there are several regions endemic for plague in our country.

Comment 3: However, this is not the universal situation and I would argue that regulatory bodies in the West have had a more cautious approach to licensing live vaccines for plague, so far at least. So, whilst I accept that 'there has been an overwhelming dominance of sub-unit vaccine' reports, (Abstract lines 13-15) these were driven by the requirements from the regulators in the West (predominantly the FDA) to 'better define' vaccines, above all with respect to absolute safety. Indeed, a microencapsulated sub-unit vaccine for plague has been licensed in Russia (referenced) but there is no elaboration of its utility. I suggest it would be useful acknowledge that a) subunit vaccines have a place in a non-endemic setting and b) that their development has helped to define factors essential for protection. Line 179 would seem an appropriate place to make such amendments.

Response 3: Thank you for your comment. The corresponding correction has been made to the text of the manuscript:

“However, it would be useful to acknowledge (i) that subunit vaccines have a place in a non-endemic setting, (ii) they may be especially useful for annual revaccinations after initial prime immunization and (iii) their development has helped to define factors essential for protection”.

Comment 4: The authors bemoan the ack of consistency in standard operating procedures to date in plague vaccine research, with which I agree and which via WHO for a in recent years is being addressed, but why do they persist in calling the pYV/pCD1 plasmid pCAD; this is a minor point, but it did rather irritate this reviewer.

Response 4: At the request of the reviewer, we have changed its name to "pYV".

Comment 5: Lines 219-220: ref 98 is quoted to support the claim that 'antibody [to plague antigens] does not correlate with protection.' This reference does not say that, rather it acknowledges that total antibody does not correlate, but that functional antibody i.e. in this case, anti-V antibody which is neutralising, does correlate. Indeed, labs around the world are starting to evaluate the development of neutralising antibody by competitive ELISA, with the hope that this may become a surrogate marker of efficacy. This is not to dismiss the role of Cell-mediated immunity, as I agree that both functional antibody and CMI are important in protection.

Response 5: A corrected version is given herewith:

“their efficacy varies significantly among non-human primate species, and total antibody titers to plague antigens do not always correlate with protection [10, 14, 98]”.

Is it OK?

Comment 6: Minor points are; line 91: a word is missing after 'attenuated'.

Response 6: There is no missing word after "attenuated" in the sentence.  «Thus, W. Kolle and R. Otto [33] showed that a single injection of attenuated in varying degrees of spontaneous Y. pestis mutants to mice, rats, guinea pigs or monkeys caused the death of some of them». The sentence is too long for English. Let me try to rephrase it.

Thus, W. Kolle and R. Otto [33] showed that a single injection to mice, rats, guinea pigs or monkeys of attenuated in varying degrees spontaneous Y. pestis mutants could cause the death of some of the animals.

Is it OK now?

Comment 7: Also please be cautious in suggesting multiple animal models are required to demonstrate vaccine efficacy (mouse, rat, guinea pig, macaque) as this adds to the difficulty of comparing data across labs/ultimately achieving a vaccine for man (with the caveat that early stage research may need several models to assess a candidate.

Line 476

Response 7: Evaluation of the efficacy of a candidate plague vaccine in several animal species (mice, rats, guinea pigs, monkeys) with various species differences in the immune system responses gives reason to hope that the vaccine will be effective in humans. However, caution should be exercised when proposing multiple animal models as this complicates the comparison of data from different laboratories.

Comment 8: Finally, I would like to have seen a conclusions/recommendation section as the bulk of the review is on EV76 and derivatives, with a late mention of a LAV plus subunit combination and other approaches to LAV's such as the delayed in vivo regulation. What is the bottom line opinion from this group of well-respected authors?    

Response 8: Conclusions

Vaccination against plague remains a challenging issue not only due to the lack of globally accepted licensed vaccines but also due to the absence of generally accepted methods for comparing their safety and efficacy. However, the plague is endemic in large populated areas of Eurasia, Africa and the Americas, which maintains its pandemic potential. It probably makes sense to continue research into development of alternative strategies for specific plague prevention based on the both live and inactivated vaccines and/or their combinations.

In our opinion, in regions endemic for plague, the most promising is prime immunization with a live vaccine based on two or three attenuated strains, at least one of which should be F1-negative. The candidate vaccine strains of the plague pathogen should not cause death in guinea pigs in doses up to 1.5×1010 cfu, and in mice in doses up to 107 cfu. The strains should have knockout mutations or, better, controlled delayed attenuation.

Then, revaccination with subunit vaccines based on F1 and V antigens in combination with bacterial ghosts or outer membrane vesicles should be carried out annually. This approach is also acceptable for the personnel of plague laboratories. If a candidate vaccine is generated that is superior to existing nationally licensed ones, it will be necessary to seek permission for its use in individual natural plague foci.

Reviewer 2 Report

Comments and Suggestions for Authors

In this review paper, authors provided comprehensive historical overview and analysis of plague vaccine development, from early empirical approaches to modern rational design strategies.

Issues:

1. manuscript is lacking clarity and well organization. Need extensive editing.

2. For the title of this manuscript, it might be better to include other forms of live vaccines against plague. Otherwise, authors may restrict Y. pestis.

3. line 72, it is confusing.

4. line 105, don’t know what the waste products are.

5. lines 131-140, the descriptions are not much related with live plague vaccines.

6.lines 154-157, descriptions are not clear.

7. lines 185-188, descriptions are not so related to the main theme of this review.

8. lines 190-209, descriptions do not fit this section.

9. a whole structure of this manuscript is not well clear, authors need to improve this.

Comments on the Quality of English Language

Extensive editing is required. Clarify needs to be largely improved.

Author Response

Dear Sir/Madam,

Thank you for your high appreciation of our manuscript.

In this review paper, authors provided comprehensive historical overview and analysis of plague vaccine development, from early empirical approaches to modern rational design strategies.

 I completely agree with you.

Issues:

Comment 1: manuscript is lacking clarity and well organization. Need extensive editing.

Response 1: Your opinion contradicts the views of the four other reviewers.

Comment 2: For the title of this manuscript, it might be better to include other forms of live vaccines against plague. Otherwise, authors may restrict Y. pestis.

Response 2: Our manuscript briefly mentions live viral- or bacterial-vectored plague vaccines (lines 181-182), so this comment is not relevant.

Comment 3: line 72, it is confusing.

Response 3: What exactly confused you on line 72?

Comment 4: line 105, don’t know what the waste products are.

Response 4: Thank you for this comment. "culture supernatant" would be more appropriate.

Comment 5: lines 131-140, the descriptions are not much related with live plague vaccines.

Response 5: When generating defined attenuated strains, it is desirable to know which gene expression switches off will not lead to a decrease in protective potency.

Comment 6: lines 154-157, descriptions are not clear.

Response 6: Why the descriptions are clear to the other four reviewers?

Comment 7: lines 185-188, descriptions are not so related to the main theme of this review.

Response 7: The text has been changed in accordance with the comment of the first reviewer.

Comment 8: lines 190-209, descriptions do not fit this section.

Response 8: Your estimation contradicts the views of the four other reviewers.

Comment 9: a whole structure of this manuscript is not well clear, authors need to improve this.

Response 9: Your judgement contradicts the views of the four other reviewers.

Comments on the Quality of English Language

Comment 10: Extensive editing is required. Clarify needs to be largely improved.

Response 10: Four of the five reviewers think that my way of expressing thoughts in English is quite understandable.

Reviewer 3 Report

Comments and Suggestions for Authors

This is a new manuscript that provides a comprehensive review of research on live attenuated vaccines against Yersinia pestis. The authors provide an interesting historical overview of the discovery and early use of live attenuated plague vaccines, followed by a detailed discussion of efficacy of EV76 and derivatives in people. In addition, there are two tables summarizing the mutant Y. pestis strains that have been investigated as candidate vaccines. Profiles for safety and efficacy are described as well as potential improvements for future development. Overall the review provides an interesting synopsis of the field and prospects for the future. However there are some areas where additional information or clarification is needed as detailed below.

1.        There are a number of live attenuated platforms that are based on Y. pseudotuberculosis and it is not clear why these were omitted from the review. It would be interesting and relevant to go over safety and efficacy of these and their potential to meet WHO standards or to be improved for future use.

2.        Table 1: could the Y. pestis biovar information be added to provide more context on the strains, whose names are largely not familiar.

3.        Table 1: MP23 is the only strain where a relative comparison to EV76 protection was not provided. Could information be added?

4.        Table 1: is dosing information available?

5.        Table 1 and 2: are there differences in the vaccine preparations and/or method of delivery that may be relevant to include in these tables?

6.        Table 2: ΔailC – there is no data entered except for one column (guinea pig protection) which says “ND”. It is not clear why this strain is included in the table nor what is known about safety and efficacy.

7.        Section 4.4 (selective potency) – it would be helpful to have a perspective on the mutant alleles that are discussed in this section that may cause a blunted response to the vaccine, as this is an interesting section that could be expanded. For example, is information available on the relative frequencies of these alleles in the population or within the trial where they were identified? How strong is the data suggesting these alleles cause reduced responsiveness?

8.        Lines 396-397 – can the authors describe the process for “testicular passages” in guinea pigs?

9.        Line 444 – provide a reference for this statement.

10.   Lines 465-467 - Recent papers by Sun and others have examined heterologous prime-boost regimens with live attenuated platforms and rLcrV. Perhaps the discussion of heterologous vaccine approaches could be expanded.

Minor:

1.        Table 1 title: it would be more precise to provide the years that are covered in Table 1 or to say “first generation” live attenuated vaccines, rather than to say “the first” strains used as vaccines.

2.        Line 314 – delete “.systems” as this appears to be an editing error.

3.        Line 316 – formatting error (“natural strains” is not italicized like the rest of the title)

4.        Line 357-359 – this is a sentence fragment and should be corrected.

5.         Lines 385-386 – this is repetitive and could be deleted.

6.        Line 390 – 397 – should this be indented as it is for numbers 1 and 3?

7.        Line 421 – delete “.” before “caused” as this should be a single sentence (as written)

8.        Line 441 – Spell out “38” when it is the first word in the sentence.

9.        Line 482 – it is unclear why reference 161 is inserted in the last sentence as it does not seem appropriate to the opinion that is expressed.

10.   Reference 4 – is this incomplete (“206 p.”)?

Author Response

Dear Sir/Madam,

Thank you for your thorough reviewing and high appreciation of our manuscript.

Comment 1: There are a number of live attenuated platforms that are based on Y. pseudotuberculosis and it is not clear why these were omitted from the review. It would be interesting and relevant to go over safety and efficacy of these and their potential to meet WHO standards or to be improved for future use.

Response 1: We believe that plague vaccines based on attenuated strains of Y. pseudotuberculosis could be the subject of a separate review, which we plan to prepare next year.

Comment 2: Table 1: could the Y. pestis biovar information be added to provide more context on the strains, whose names are largely not familiar.

Response 2: Unfortunately, these data are not available in the cited literature.

Comment 3: Table 1: MP23 is the only strain where a relative comparison to EV76 protection was not provided. Could information be added?

Response 3: Unfortunately, this information is not available in the cited literature.

Comment 4: Table 1: is dosing information available?

Response 4: The dosing information is also unavailable.

Comment 5: Table 1 and 2: are there differences in the vaccine preparations and/or method of delivery that may be relevant to include in these tables?

Response 5: The main historical information is taken from the book by E.I. Korobkova (1956). Unfortunately, there is very few relevant information.

Comment 6: Table 2: ΔailC – there is no data entered except for one column (guinea pig protection) which says “ND”. It is not clear why this strain is included in the table nor what is known about safety and efficacy.

Response 6: Thank you for your attention. We have no published data on this ΔailC strain. We will remove this line from the table.

Comment 7: Section 4.4 (selective potency) – it would be helpful to have a perspective on the mutant alleles that are discussed in this section that may cause a blunted response to the vaccine, as this is an interesting section that could be expanded. For example, is information available on the relative frequencies of these alleles in the population or within the trial where they were identified? How strong is the data suggesting these alleles cause reduced responsiveness?

Response 7: Unfortunately, due to the MDPI's strong recommendations, this section has been shortened. The original article is published in English (https://www.elibrary.ru/item.asp?id=29455131). Unfortunately, the number of volunteers in the first phase of clinical trials was limited and the data obtained are not very reliable.

Comment 8: Lines 396-397 – can the authors describe the process for “testicular passages” in guinea pigs?

Response 8: Previously, testicular passages have been described quite often in English-language literature, where they can be found online. We can provide access data to a Russian-language patent (Patent RU #2510825. Sposob polucheniya preparata na osnove vaktsinnogo shtamma chumnogo mik-roba [Method of obtaining preparation based on vaccine strain of plague microbe]: 2012, Publ. 10.04.2014, Bull. no. 10. https://www.elibrary.ru/item.asp?id=37796412), which can be translated into English online (https://translate.google.com/details?sl=en&tl=ru&op=translate).

However, in our time, this inhumane method is unlikely to be approved by the bioethics commission.

Comment 9: Line 444 – provide a reference for this statement.

Response 9: The link to the original source is inserted on line 463.

Comment 10: Lines 465-467 - Recent papers by Sun and others have examined heterologous prime-boost regimens with live attenuated platforms and rLcrV. Perhaps the discussion of heterologous vaccine approaches could be expanded.

Response 10: We are currently planning a series of experiments on heterogeneous immunization, and we think it would be appropriate to discuss the questions you suggest in an experimental article that will be published based on materials of our experiments.

Comment 11: Table 1 title: it would be more precise to provide the years that are covered in Table 1 or to say “first generation” live attenuated vaccines, rather than to say “the first” strains used as vaccines.

Response 11: Thank you for your comment, the correction has been made to the text.

Comment 12: Line 314 – delete “.systems” as this appears to be an editing error.

Response 12: Thank you for your comment, the correction has been made to the text.

Comment 13: Line 316 – formatting error (“natural strains” is not italicized like the rest of the title)

Response 13: Corrected.

Comment 14: Line 357-359 – this is a sentence fragment and should be corrected.

Response 14: Your recommendation cannot be implemented because the paraphrasing carried out at the urgent request of MDPI changed the arrangement of words and the words themselves in the text fragment that includes the section of text you proposed for paraphrasing.

Comment 15: Lines 385-386 – this is repetitive and could be deleted.

Response 15: Strikethrough in the main text. Thanks again for your attention.

Comment 16: Line 390 – 397 – should this be indented as it is for numbers 1 and 3?

Response 16: No, there is no need for that, since all three paragraphs are dedicated to animalization.

Comment 17: Line 421 – delete “.” before “caused” as this should be a single sentence (as written)

Response 17: Deleted.

Comment 18: Line 441 – Spell out “38” when it is the first word in the sentence.

Response 18: Done.

Comment 19: Line 482 – it is unclear why reference 161 is inserted in the last sentence as it does not seem appropriate to the opinion that is expressed.

Response 19: Done.

Comment 20: Reference 4 – is this incomplete (“206 p.”)?

Response 20: This citation is based on samples we found on the internet. We found no other information.

Reviewer 4 Report

Comments and Suggestions for Authors

This review by Andrey P. Anisimov et.al. is very interesting and informative. I believe it will be valuable for researchers in the field of plague study. Some minor comments are outlined below.

Line 171, Table 2,Live attenuated Y. pestis strains used as vaccines against plague and generated using molecular bacteriology methods. To my knowledge, a live attenuated vaccine based on EV76 has been reported in 2024 (PMID: 38547321), which has not been included in Table 2.  

Lines 280-293, the authors listed 7 criteria for selecting candidate vaccine strains. The live attenuated strains of Y. pestis could possibly loss one or several genes that are critical for virulence in mice or other hosts. Once loss of these genes, the mutants might exhibit different phenotypes that are very different from the wild-type strains. I do not suggest to propose criteria that are very specific and restrict the strains’ phenotypes in a narrow scale, because some candidate vaccine strains could not fulfill some of these criteria.

Line 292 to 203, “pFra (60 MDa), pCad (47 MDa) and pPst (6 MDa)”, I would like to kindly recommend the authors to consider using the prevalence nomenclature when referring these plasmids. For instance., it’s better to use pMT1, instead of pFra, pCD1 instead of pCad, pPCP1 instead of pPst. It should be easier for readers to follow the context of the manuscript.

Line314, please remove the repeated word “systems”.

Author Response

Dear reviewer,

Thank you for taking the time to review our paper, provide comments to help to improve it and high appreciation of our manuscript.

Comment 1: Line 171, Table 2,Live attenuated Y. pestis strains used as vaccines against plague and generated using molecular bacteriology methods. To my knowledge, a live attenuated vaccine based on EV76 has been reported in 2024 (PMID: 38547321), which has not been included in Table 2.

Response 1: Changes made to the table based on the reviewer's recommendation are highlighted in red.

Table 2. Live attenuated Y. pestis strains used as vaccines against plague and generated using molecular bacteriology methods [54-63].

Y. pestis strain

Virulence for (approximate LD50 values in cfu)

Protective potency for

mice

guinea pigs

mice

guinea pigs

subcutaneous challenge

Biovar Microtus strain 201

Avirulent to humans or primates

201

3.1

ND

+

+

201Δyp1

4.8×104

ND

+

ND

201Δyp2

9.0×104

ND

+

ND

201Δyp1&yp2

5.2×104

ND

+

ND

Wild type subspecies pestis

< 10

< 10

+

+

Δpgm

> 6.3×107 CFU

> 1.5×1010

+

+

pMT1-

< 10 to > 108

< 10 to > 108

ND

ND

pYV-

> 1.0×108

> 1.5×1010

-

ND

pMT1-

< 10

< 10

+**

ND

Δpgm pMT1-

> 1.0×108

ND

+

ND

Δpgm Δyp1

> 1.0×108

ND

ND

ND

Δpgm Δyp2

> 1.0×108

ND

ND

ND

Δpgm Δyp1&yp2

> 1.0×108

ND

ND

ND

Δpla

< 10 to > 108

< 10

+

ND

ΔnlpD

> 107

> 1.5×1010

+

-

ΔyopH

> 107

ND

±

ND

Δdam

2.3×103

ND

+

ND

ΔrelA ΔspoT

5.8×105

ND

+

ND

Δcrp

> 3×107

ND

+

ND

ΔyscB

> 106

ND

+

ND

ΔglnALG

> 105

> 107

+

+

ΔmetQ

> 105

> 108

-

-

Δlpp ΔmsbB Δail

> 2×106

ND

+

ND

Δlpp ΔmsbB::ailL2

> 2×106

ND

+

ND

Δypo2720-2733Δhcp3

60%*

ND

+

ND

ΔvasKΔhcp6

60%*

ND

±

ND

∆yscN

> 3.2×107

ND

+

ND

∆surA

> 2.1 × 105 

ND

+

ND

intranasal challenge

Δlpp ΔmsbB Δpla

> 2×106

ND

+

ND

ΔsmpB-ssrA

> 106

ND

+

ND

*% Animal survival in pneumonic plague challenge; ND, no data.

**+ – protective from F1-negative strains

Comment 2: Lines 280-293, the authors listed 7 criteria for selecting candidate vaccine strains. The live attenuated strains of Y. pestis could possibly loss one or several genes that are critical for virulence in mice or other hosts. Once loss of these genes, the mutants might exhibit different phenotypes that are very different from the wild-type strains. I do not suggest to propose criteria that are very specific and restrict the strains’ phenotypes in a narrow scale, because some candidate vaccine strains could not fulfill some of these criteria.

Response 2: You are absolutely right and the national criteria for selecting Yersinia pestis vaccine strains in Russia are morally obsolete.In our opinion presented on lines 298-313, the number of criteria should be reduced, it is advisable to set two main ones: safety and high protective activity.In other parameters, strains may differ from the seven requirements stated in the guidelines.

Comment 3: Line 292 to 203, “pFra (60 MDa), pCad (47 MDa) and pPst (6 MDa)”, I would like to kindly recommend the authors to consider using the prevalence nomenclature when referring these plasmids. For instance., it’s better to use pMT1, instead of pFra, pCD1 instead of pCad, pPCP1 instead of pPst. It should be easier for readers to follow the context of the manuscript.

Response 3: The text now includes the substitutions pMT1 instead of pFra, pPCP1 instead of pPst; the substitution pYV instead of pCad was made earlier at the suggestion of one of the previous reviewers.

Comment 4: Line314, please remove the repeated word “systems”.

Response 4: Removed.

Reviewer 5 Report

Comments and Suggestions for Authors

Some of the writing contains long run-on sentences.  These could be modified without loss of meaning. The major ones I think would benefit include:

The first sentence (lines 49-53) in section 2.

Lines 64-69 spanning six lines of type

Lines 477-480--too long

Line 62--change 'but' to 'so', i.e. The dead microbes were unable to cause an infectious process, so......

In the discussion of virulence determinants, lines 149-159 the authors discuss Burrow's prophetic recognition of virulence factors, the loss of which would attenuate or eliminate virulence.  This is really the basis of Stan Falkow's Molecular Koch's postulates that were verified with the Y. pseudotuberculosis invasin gene.  Would it not enhance this paragraph to cite his article?

Lines 185-189 deal with subunit vaccines where the authors state they have "no or limited innate defense triggers".  The incorporation of Toll-like receptor agonists in subunit vaccines are addressing this problem---I think this should be mentioned--either with or without references.

Line 314 stating ".....belongs to cellular factors of the immune system systems" is awkward.  Is this a typos

The sentence beginning with line 445 is awkward.  I suggest changing to "To further optimize the EV vaccine one research group has proposed deletion of the pla gene encoding the plasminogen activator."  Further, "It is likely that the deletion of other ballast components........"

Sentence starting with line 451 Taking into account the poly-host nature, the several clinical formsof plague, and the multi-- include 'the' in front of several.

I love the admonition in line 481 recommending readers read Roy Curtiss's papers!  But why is reference 161 cited after "They deserve it"???

These are minor suggestions.  Overall the review is well written and timely coming from a well-respected Yersinia group.  Thanks for bringing this all together for the rest of us.

Author Response

Dear reviewer,

Thank you for taking the time to review our paper, provide comments to help to improve it and high appreciation of our manuscript.

Some of the writing contains long run-on sentences.  These could be modified without loss of meaning. The major ones I think would benefit include:

Comment 1: The first sentence (lines 49-53) in section 2.

Response 1: Done

Comment 2: Lines 64-69 spanning six lines of type

Response 2: Done

Comment 3: Lines 477-480--too long

Response 3: Done

Comment 4: Line 62--change 'but' to 'so', i.e. The dead microbes were unable to cause an infectious process, so......

Response 4: Done

Comment 5: In the discussion of virulence determinants, lines 149-159 the authors discuss Burrow's prophetic recognition of virulence factors, the loss of which would attenuate or eliminate virulence.  This is really the basis of Stan Falkow's Molecular Koch's postulates that were verified with the Y. pseudotuberculosis invasin gene.  Would it not enhance this paragraph to cite his article?

Response 5: Done.

Comment 6: Lines 185-189 deal with subunit vaccines where the authors state they have "no or limited innate defense triggers".  The incorporation of Toll-like receptor agonists in subunit vaccines are addressing this problem---I think this should be mentioned--either with or without references.

Response 6: Done.

Comment 7: Line 314 stating ".....belongs to cellular factors of the immune system systems" is awkward.  Is this a typos

Response 7: Yes, it is. We corrected this typo.

Comment 8: The sentence beginning with line 445 is awkward.  I suggest changing to "To further optimize the EV vaccine one research group has proposed deletion of the pla gene encoding the plasminogen activator."  Further, "It is likely that the deletion of other ballast components........"

Response 8: Done.

Comment 9: Sentence starting with line 451 Taking into account the poly-host nature, the several clinical formsof plague, and the multi-- include 'the' in front of several.

Response 9: Done.

Comment 10: I love the admonition in line 481 recommending readers read Roy Curtiss's papers!  But why is reference 161 cited after "They deserve it"???

Response 10: Unnecessary quotation has been removed.

Comment 11: These are minor suggestions.  Overall the review is well written and timely coming from a well-respected Yersinia group.  Thanks for bringing this all together for the rest of us.

Response 11: Thank you for your high rating.

Round 2

Reviewer 2 Report

Comments and Suggestions for Authors

The manuscript was improved after revision. There are some issues which still need to be addressed.

Issues:

1. lines 49-79, it is better to briefly and concisely describe the two paragraphs.

2. “An important step in the generation of attenuated strains is cloning” is confusing.

3. line 80, “A. Yersin” should be “Alexandre Yersin”

4. line 84, “plague microbes” is not proper.

5. lines 94-96 and lines 89-90 are repeated.

6. line 98, “a rule avirulent…” is not proper.

7. line 110, “effectiveness” should be “efficacy”

8. line 175, “plague pathogenesis” should be “Y. pestis pathogenesis”

9. lines190-193, the description is not accurate. Nanoparticle plague vaccines seem to show good immunogenicity.

10. line 312, “high vaccine protectivity” should be “high vaccine protective efficacy”

11. lines353-362, no descriptions abut live plague vaccines which are main points in this review.

12. line 435, “smaller” should be “less”.

13. English editing is still required to improve clarity.

Comments on the Quality of English Language

English editing is required

Author Response

We are sincerely grateful to our reviewer for continuing to improve our manuscript.

Comment 1: lines 49-79, it is better to briefly and concisely describe the two paragraphs.

Response 1: The text has been shortened.

Comment 2: (72-73) “An important step in the generation of attenuated strains is cloning” is confusing. 

Response 2: Corrected and shortened.

Comment 3: line 80, “A. Yersin” should be “Alexandre Yersin”

Response 3: Corrected.

Comment 4: line 84, “plague microbes” is not proper.

Response 4: Corrected.

Comment 5: lines 94-96 and lines 89-90 are repeated.

Response 5: Duplication removed.

Comment 6: line 98, “a rule avirulent…” is not proper.

Response 6: Corrected.

Comment 7: line 110, “effectiveness” should be “efficacy”

Response 7: Corrected.

Comment 8: line 175, “plague” should be “Y. pestis pathogenesis”

Response 8: I cannot agree with our reviewer.

The word "pathogenesis" means the development of a disease, while "pathogenicity" means the property of causing a disease. Thus, the correct phrases are plague pathogenesis and Y. pestis pathogenicity.

Comment 9: lines190-193, the description is not accurate. Nanoparticle plague vaccines seem to show good immunogenicity.

Response 9: The text on these lines refers not so much to nanoparticle vaccines as to all low-component vaccines. I have modified the sentence. «However, according to a number of researchers, they have no or limited innate defense triggers and do not guarantee the formation of a good immune memory».

Comment 10: line 312, “high vaccine protectivity” should be “high vaccine protective efficacy”

Response 10: Corrected.

Comment 11: lines353-362, no descriptions abut live plague vaccines which are main points in this review.

Response 11: Here we are writing about antigens common to the both live and inactivated plague vaccines.

Comment 12: line 435, “smaller” should be “less”.

Response 12: Corrected.

Comment 13: English editing is still required to improve clarity. 

Response 13: As recommended by the Reviewer, the English has been additionally checked for clarity and several corrections have been made.